# The Impact of the COVID-19 Pandemic on Fatal Road-Traffic Accidents: A Five-Year Study on Medicolegal Autopsies in Timis County, Romania

Ştefania Ungureanu [1,2,3,4], Veronica Ciocan [2,3,4,*], Camelia-Oana Mureşan [2,3,4], Emanuela Stan [1,2,3,4], Georgiana-Denisa Gavriliţă [1,3,4], Alexandra Sirmon [5,6], Cristian Pop [7] and Alexandra Enache [2,3,4]

1. Doctoral School, Victor Babes University of Medicine and Pharmacy Timisoara, 2 Eftimie Murgu Square, 300041 Timisoara, Romania; stefania.ungureanu@umft.ro (Ş.U.); emanuela.stan@umft.ro (E.S.); denisa.tincu@umft.ro (G.-D.G.)
2. Discipline of Forensic Medicine, Bioethics, Deontology and Medical Law, Department of Neuroscience, Victor Babes University of Medicine and Pharmacy Timisoara, 2 Eftimie Murgu Square, 300041 Timisoara, Romania; muresan.camelia@umft.ro (C.-O.M.); enache.alexandra@umft.ro (A.E.)
3. Timisoara Institute of Legal Medicine, 1A Ciresului Street, 300610 Timisoara, Romania
4. Ethics and Human Identification Research Center, Department of Neuroscience, Discipline of Forensic Medicine, Bioethics, Deontology and Medical Law, Victor Babes University of Medicine and Pharmacy Timisoara, 2 Eftimie Murgu Square, 300041 Timisoara, Romania
5. Pius Branzeu Emergency County Clinical Hospital Timisoara, 156 Liviu Rebreanu Bld., 300723 Timisoara, Romania; alexandra.sirmon@gmail.com
6. Residency Program in Epidemiology, Victor Babes University of Medicine and Pharmacy Timisoara, 2 Eftimie Murgu Square, 300041 Timisoara, Romania
7. Department of Mechatronics, Faculty of Mechanical Engineering, University Politehnica Timisoara, 1 Mihai Viteazu Bld., 300222 Timisoara, Romania; cristian.pop@upt.ro
* Correspondence: veronica.ciocan@umft.ro; Tel.: +40-722944453

**Abstract:** Road traffic accidents (RTAs) represent the key sign of the level of road safety. Romania once held the record for road deaths among European Union (EU) countries and as of 2023, it came second place. It is of utmost importance to assess whether measures that restricted human mobility during the coronavirus disease 2019 (COVID-19) pandemic led to a significant reduction in road fatalities. This study assesses the impact of the COVID-19 pandemic on victims of fatal RTAs by analyzing medicolegal autopsies from the Timisoara Institute of Legal Medicine (TILM), Timis County, Romania. Materials and methods: Medicolegal autopsy records of RTA victims from TILM in a 5-year period (2017–2021) were analyzed. Results: 395 cases (10.5%) were represented by victims of fatal RTAs. The reduction in the number of cases in the pandemic period was not statistically significant ($p = 0.061$) compared to the pre-pandemic period, but the number of victims of RTAs decreased by 17.6%. This highlights the importance of understanding the role of other risk factors in fatal RTAs, since a lesser volume of traffic did not cause a significant decrease in road fatalities. Male victims were predominant, with 18–50 years being the most affected age group. In the pandemic period, the most affected age groups were 31–40 (18.5%), followed closely by 41–50 (17.6%) and 18–30 (16.7%). In the pre-pandemic period, the first place was held by people in the age interval of 61–70 (20.5%), followed by 18–30 (19.2%). Drivers were the most involved type of road user, and a slight increase in the proportion of cyclists (13.9% from 10.5%) and motorcyclists (6.5% from 5.9%) was noted. Conclusions: Our findings show that measures implemented to control the COVID-19 pandemic may have had a positive effect on the reduction of RTAs, as shown by the information based on medicolegal autopsies in Timis County, Romania, but more attention needs to be focused on other risk factors. Further studies need to identify reasons for the small reduction in fatal injuries when the volume of traffic was reduced during mandated national lockdown.

**Keywords:** fatal road traffic accidents; medicolegal autopsy; COVID-19 pandemic; lockdown

## 1. Introduction

Road traffic accidents (RTAs) are a crucial and recognized international problem. They represent a significant public health danger regarding injuries, incapacitations, and deaths [1]. According to the World Health Organization (WHO), roughly 1.19 million individuals died in 2021, a 5% decline since 2010 [2]. As of 2019, RTAs remained the twelfth leading cause of death for individuals regardless of age, while in 2016, they represented the eighth leading cause of death. However, RTAs still represent the main cause of death for children and young people aged 5–29 years [1,2].

Each year, 127,000 people die from road crashes in the WHO European Region [3]. United Nations Economic Commission for Europe (UNECE) data illustrate that in 2019, about 98,500 individuals died because of RTAs, with a median of 270 people dying daily from road accidents in the area [4].

In 2023, Romania occupied the second place in the European Union (EU) countries in road fatalities, with an average of 81 deaths per 1 million people. This, however, is seen as an improvement from 2022, when Romania was the first in road deaths with 86 deaths/1 million people, with the EU average being 46 deaths/1 million people. According to Romanian police statistics, among risk factors for severe RTAs in 2019–2023, we see excessive speeding (16%), illegal road crossing for pedestrians (13%), bike user-related risk factors (11%), and drivers not adhering to road rules relating to pedestrian crossings or other vehicles (17%) [5,6].

Traffic accidents represent the key sign of the degree of road safety. In developed and developing nations, road safety matters represent a great concern to reduce the number of RTAs that happen [7]. It is mandatory to examine RTA deaths to implement measures to decrease them [8]. The human factor is the main factor responsible for RTAs, as road users are not complying with the rules of the road [9]. Human factors include reckless driving, driving while inebriated, drug-induced impairment of driving, inexperienced drivers, over-speeding, driving while sleepless and weary, and using a mobile phone [10,11]. The second factor that needs to be considered is the vehicle factor, including bad tire control, inadequate lighting, breakdown of engines and gearboxes, and weak brakes. The expansion in vehicle numbers and urbanization progression have contributed to traffic congestion in cities and subsequently to the event of more RTAs [11]. Traffic jams are also one of the main causes that produce road accidents [7].

The year 2020 will be considered for the coronavirus disease 2019 (COVID-19) pandemic, which has touched several million individuals (confirmed cases) worldwide, leading to hundreds of thousands of fatalities [12]. COVID-19 was declared a pandemic by the WHO on 11 March 2020 [13]. Infectious pandemics substantially and precisely alter a community's health and budget [14]. The COVID-19 pandemic has substantially impacted all aspects of life, and its destructive results will vibrate throughout the coming years [8]. Countries began employing diverse tactics to decrease the spread of the pandemic, such as physical distancing, closure of schools, stay-at-home orders, quarantine, and restrictions on mass gatherings and travel [14]. Several national governments reacted by enforcing lockdowns to decrease the dissemination of infection and possible fatalities [12]. The various lockdown measures have been reported to have had an important effect on different types of traumas [15]. One of the limitations proposed restricted individual mobility, seeing that air and land transportation represented two of the key directions for the spread of COVID-19 [12]. The spread of COVID-19 may have been increased by human mobility [16]. In Italy, researchers illustrated an explicit positive association between the cases of COVID-19 each day with travels made during the three weeks before. Moreover, transport accessibility had the highest contribution in explaining the incidence of infected cases [17,18].

Several researchers have analyzed the complicated relationship between the COVID-19 pandemic, lockdown measures, human mobility patterns, and RTAs. In the next paragraphs, we review the main studies on the matter.

Traffic-related incidents reduced substantially with stay-at-home measures, which constrained human movement [19]. Vehicle mobility was significantly lowered by more

than 50% internationally, with a reduction of 50% to 60% in the Asian nations and up to 80% in the European ones [14]. The limitations on movement established during the COVID-19 outbreak indicated a severe decrease in road transportation, resulting in the absence of traffic jams, which affected road traffic collisions [7,12,14]. The decline of traffic overcrowding contrasted in numerous countries, ranging between 25% and 75% during lockdown in nations in Europe, with the greatest reduction, of 75%, in Spain, France, and Italy in April 2020 in comparison to April 2019. Traffic was reduced by 40% in the United States of America, 63% in the United Kingdom, 77% in South Africa, and 43% in Australia at the same time in comparison to the month of April 2019 [14].

Moreover, the restrictions imposed changed the way people chose to travel. The Boston Consulting Group, by studying 5000 city inhabitants in the United States, Europe, and China in April 2020, illustrated that private modes of transport (walking, scooters, and bikes) increased in all three regions due to the low possibility of contracting the virus [20]. Apple showed changes in human mobility worldwide by using the mobility data of mobile phone operators. In the United States, United Kingdom, Italy, Australia, Brazil, and India, a mobility reduction in March–May 2020, with a meaningful decline in April, was noted [16]. In Australia, an 80% decrease in public transport use was noted, with an increased tendency for cyclists and car use [21]. In Switzerland, researchers assessed a drop between 60 and 95% in everyday travel distance using a car and public transportation because of national restrictions [22]. In Germany, structural changes in mobility patterns were noted (i.e., more regional travel, variations in the smallest distance, and a drop in long-distance travel) [23]. While investigating the influence of restrictions and individual movement on COVID-19 fatalities in the United Kingdom, a steady reduction was found in human motion in the month of May 2020, which is positively associated with the drop in the number of fatalities [24]. In Greece, a reduction of 50% in everyday travel per individual because of the national lockdown was noted [25]. In France, researchers observed that lockdown decreased short- and long-range movement throughout the nation by up to 65% [26]. In Northern Ireland, a reduction of 26% in admission to hospital was noted in the lockdown period, in comparison to the same time in 2019, and a large proportion of that decrease was because of fewer RTAs. This was also confirmed by the police road deaths and injury statistics. The months of lockdown (March, April, and May) experienced reductions in people killed or severely injured on the road. They estimated that traffic levels over-halved during lockdown, which could be the reason for the reduction in victims [27].

In Turkey, national measures restricting mobility began on 12 March 2020, with schools and universities closing, non-essential businesses suspended (bars, night clubs, museums, hairdressers, and barbershops), curfew for people aged 65 and older and younger than 20, and on 3 April, a quarantine was imposed on 30 metropolitan cities. This culminated in a curfew for 48 h for everyone. Umut Oguzoglu analyzed the results regarding traffic accidents and related injuries and road deaths due to restrictions imposed by the government to fight COVID-19 cases in Turkey. In the month of April in 2020, when the strictest measures were taken for the entire month, RTAs declined by almost 60%, and deaths dropped by 43%, while injuries dropped by 64% compared to April 2019. These are added to the decline in March 2020 when milder restrictions were in order during half of the month [28].

In Sydney, Australia, positive outcomes have been noted, with a discernible drop in the number of overall crashes and the frequency of vehicle breakdowns during lockdown periods. When comparing the number of fatalities reported for April 2020 with the same period in 2019 and 2018, there was an average 30% decrease [19].

Because of the safeguards put in place as a result of the pandemic, there have been fewer traffic mishaps, injuries, and fatalities in Japan. During the peak months of COVID-19, road deaths registered a descending tendency nationwide, but the overall road deaths varied over the period, and even an increase in some prefectures was noted. This was attributed to the high density of vehicles and people. Furthermore, in some places, the decline in traffic volume contributed to a rise in traffic fatalities due to an increase in speeding. All things considered, there was a strong negative link between the number of

COVID-19 infections and the number of traffic accidents, injuries, and fatalities across the country [29].

A study that analyzed data from the Statewide Traffic Accident Records System maintained by the Missouri State Highway Patrol in investigating the effects of the mandated societal lockdown and traffic accidents found that the daily counts of RTAs decreased during the entire period. An important decrease in RTAs resulting in insignificant or no injuries was found but no significant reduction in incidents resulting in severe or deadly injuries. Potential causes could include faster-moving vehicles as a result of less congestion, which could counteract the beneficial effects of slower-moving vehicles by increasing the likelihood of serious or deadly RTAs. Other potential factors include an increase in the number of drunk and drugged drivers, modifications to road-safety awareness efforts, and a rise in the speed of heavy-vehicle traffic without a corresponding decrease in volume [30].

Seeing these global statistics related to the changes in road accidents and road deaths suffered due to restrictions implemented to slow the dissemination of the pandemic, we wanted to see how Romania was affected by the restrictive measures the Romanian government forced. On 26 February 2020, the first case of COVID-19 was reported in Romania, and during the first ten days of March, confirmed cases were reported with a frequency of zero to four cases per day, totaling up to seventeen cases [31]. With COVID-19 reaching pandemic proportions, the president declared a state of emergency nationwide for 30 days on 16 March 2020 which was extended until 14 May [32,33]. These two periods represent the lockdown that was instituted in Romania. The main measures imposed were as follows:

1.  Isolation of people who encountered infected people and those who traveled in at-risk areas.
2.  Gradual closure of frontier points.
3.  Reduced or forbidden human and vehicle mobility in specific areas in the country or within a certain time frame.
4.  Gradual forbiddance of all transport by car, train, boat, or plane.
5.  Temporarily closure of restaurants, hotels, coffee places, nightclubs, casinos, and other public places.
6.  Assurance of protection for workers in essential businesses such as the power, water, and natural gas industries.
7.  Personal protection equipment dissemination among citizens, with identifying suppliers and distributors, as well as medication and sanitizers.
8.  Public hospital limitation to only emergency cases (including COVID-19) [32,33].

Additionally, all schools were closed, and online teaching represented the new normal. Non-essential workers started working from home and online meetings replaced the ones face-to-face. Moreover, a permission slip was instituted for people who needed to leave their homes for buying groceries, procuring medication, going to work for essential workers, or providing those in need with essential things [32,33].

A positive outcome of the measures implemented to curb the transmission of COVID-19 was the decline in traffic incidents on urban and interurban highways; this led to a pronounced fall in traffic-related injuries and deaths [12]. This was in accordance with the European Transport Safety Council statistics that showed that the restrictions that were implemented in Romania in the declared state of emergency led to a reduction of 52% in the number of road fatalities in April 2020 in comparison to the average of the same months in 2017–2019 [34]. Seeing these national statistics, we wanted to analyze the impact of the COVID-19 pandemic and the imposed restrictions on the number of road fatalities in a particular region in Romania by studying road victims over a period of five years and making a comparison between the pandemic period and the previous years. We expected the number of road deaths to be reduced in comparison to previous years due to reduced mobility in the context of restrictive measures for containing the spread of the pandemic.

This study aimed to evaluate the impact of the COVID-19 pandemic and the imposed restrictions on the number of victims of fatal RTAs in Timis County in Western Romania by

analyzing data obtained from medicolegal autopsy records from the Timisoara Institute of Legal Medicine (TILM) in a 5-year period (1 January 2017–31 December 2021).

It is well known that medicolegal autopsies are essential for the epidemiological assessment of RTA cases [8]. They facilitate establishing the exact cause and manner of death, time since death, and circumstances of death [35]. In Romania, medicolegal autopsies are mandatory in violent, suspicious deaths or when the cause of death is unknown. This applies to all road fatalities, forgetting the time since the accident and the victim's death [36].

To our knowledge, this research is the first study that addresses the impact of the COVID-19 pandemic on RTAs in the Western part of Romania by assessing medicolegal autopsies, and it will bring new proof to the increasing research on the pandemic's numerous consequences on global transportation by adding a new country to the list.

Recognizing traffic trauma and their admission models throughout lockdowns, such as that during the COVID-19 pandemic, would help medical officials to better rationalize assets during related situations [37].

Seeing that Romania held the record for road deaths in the EU and, as of 2023, occupies the second place, we think it is quite crucial to assess whether road fatalities in Romania were influenced by the measures imposed during the pandemic and to see if restrictive patterns of human and vehicular movements may reduce fatal RTAs. This is needed to develop new information and public campaigns targeted at decreasing RTAs and road deaths. Moreover, with this research comes the need to address all the risk factors that lead to fatal RTAs, such as reckless driving, driving while intoxicated, tired driving, driving and disregarding the road, disregarding the traffic laws, risk factors associated with other types of road users (pedestrians, bikers, or motorcyclists), the need for protective equipment (child protective means, seat belts, and helmets), walking while intoxicated, and risk factors that address the vehicle and the road infrastructure.

## 2. Materials and Methods

This study analyzes the impact of the COVID-19 pandemic and the restrictive measures the government imposed to stop the spread of the virus on the number of victims of fatal RTAs in Timis County, Romania, by assessing medicolegal autopsy records from TILM over a 5-year period (2017–2021).

A record-based, retrospective, analytical study was conducted using medicolegal autopsy records of all RTA victims at TILM in a 5-year period (2017–2021). The cases reported from 1 January 2017 to 31 December 2021 were retrieved and grouped according to the victims' demographics (age and gender), the date of the accident, and the type of road user (driver, passenger, pedestrian, cyclist, motorcyclist, or scooter). This research follows a two-stage study to assess the COVID-19 pandemic impact and the mandated restrictive measures on victims of fatal RTAs. In the first stage, data was organized into two groups; the first group was termed the pre-pandemic period (1 January 2017–10 March 2020); the second group was termed the pandemic period (11 March 2020–31 December 2021). In the second stage of the study, data were further stratified into three groups: the first group was the lockdown period (16 March–14 May 2020), the second group was the pre-lockdown period (same period during 2019), and the third group was the post-lockdown period (same period during 2021). We studied the same parameters as in the first-stage study.

This study was overseen in accordance with the Declaration of Helsinki and was approved by the Research Ethics Committee of Victor Babes University of Medicine and Pharmacy Timisoara, registered under decision number 88/19.12.2022 rev 2024.

Inclusion criteria: medicolegal autopsy records from TILM of victims of RTAs in a 5-year period (1 January 2017–31 December 2021).

Exclusion criteria: medicolegal autopsy records from TILM of people who were not victims of RTAs and records not pertaining to the period mentioned in the inclusion criteria above were excluded.

Study tool: The fatal RTAs in each group were collected and introduced into an Excel sheet and then analyzed using simple mathematical tools. Statistical analysis was

performed using IBM SPSS Statistics 27. In the statistical analyses carried out, we define statistical significance, alpha, as 0.05 for 2-sided tests. The chosen level of significance corresponds to a 95% confidence interval. Descriptive results were reported for all variables. In this regard, frequency and contingency tables were used. Continuous variables were reported as means with standard deviation (SD). Categorical variables were reported as numbers and percentages. *t*-tests, Z-tests, and Pearson Chi-square tests (for continuous and categorical variables, respectively) were appropriately used when comparison among variables was performed. Due to the relatively small size of our sample, whenever Pearson Chi-square test assumptions were not met, Fischer's exact test was used.

## 3. Results

The total number of medicolegal autopsies that were performed at TILM between 1 January 2017 and 31 December 2021 was 3752; 10.5% ($n$ = 395) of them were represented by autopsies of victims of fatal RTAs, which equaled 21.8% of all violent deaths ($n$ = 1808). We noted that three victims who were involved in fatal RTAs in 2017 died and were autopsied the following year. This aspect was also observed in 2018 (five cases) and 2019 (three cases). In the coming study, we analyzed the distribution and characteristics of victims of fatal RTAs regarding the date of the accident and not the date of the autopsy.

When the annual rate of case spread was evaluated, 2018 had the highest number of RTA deaths ($n$ = 106), followed by 2017 ($n$ = 97) and 2019 ($n$ = 73). However, in the year of the COVID-19 pandemic (2020), in comparison to 2018, their number reduced by 44.3% ($n$ = 59). In 2021, 60 cases of RTA fatalities were reported.

### 3.1. First-Stage Study: Pre-Pandemic Group Versus Pandemic Group

3.1.1. Annual Medicolegal Autopsy Records Regarding the Pre-Pandemic and the Pandemic Period

Regarding the number of fatal cases in the pre-pandemic period as opposed to the pandemic period, we noticed that in the first period, from 1 January 2017 until 10 March 2020, 11.14% ($n$ = 287) of all medicolegal autopsies during this period ($n$ = 2576) were represented by victims of fatal RTAs. The number was reduced by 17.6% ($n$ = 108, 9.18%) in the pandemic period ($n$ = 1176). These findings are illustrated in Table 1.

**Table 1.** Comparison of fatal RTAs in the pre-pandemic period versus the pandemic period as a result of a 5-year study (2017–2021) based on medicolegal autopsy records from Timisoara Institute of Legal Medicine (TILM), Romania.

| Period | Number of Medicolegal Autopsies | Number of Autopsies of Victims of Fatal RTAs | Number of Cases |
|--------|---------------------------------|---------------------------------------------|-----------------|
| Pre-pandemic period | 2576 | 287 | 288 |
| Pandemic period | 1176 | 108 | 107 |
| Total | 3752 | 395 | 395 |

However, based on the two-proportion Z-test results and the Pearson Chi-square test results, the two time periods we analyze do not statistically differ in any way that is significant (difference of 0.0195763, confidence interval of (−0.000919, 0.040072); Z-Value = 1.87, $p$ = 0.061; Pearson Chi-square = 2.00, $p$ = 0.157).

Regarding the average daily number of victims in the pre-pandemic period versus the pandemic period, we observed a drop in cases from 0.25 cases/day to 0.16 cases/day.

3.1.2. Monthly Variations of Fatal RTAs

In the pre-pandemic period, the maximum number of victims of fatal RTAs was reported in May 2018 ($n$ = 19). In comparison, in the pandemic period, in the month of May there were observed only three cases in 2020 and four cases in 2021. The lowest

number of fatal RTAs in the pandemic period was noted in October 2020 (*n* = 1), whereas in the pre-pandemic period, we found ten cases in 2017, fourteen cases in 2018, and eight cases in 2019. The monthly distribution of cases regarding the comparison between the pre-pandemic period and the pandemic period is illustrated in Figure 1.

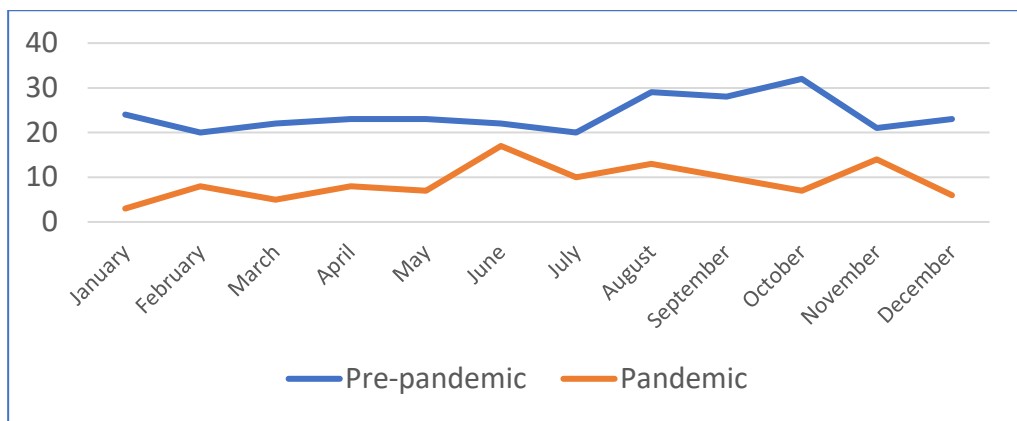

**Figure 1.** Histogram of victims of fatal RTAs according to the month of the accident in the 5-year period (2017–2021) based on medicolegal autopsy records from TILM, Romania; comparison between the pre-pandemic and the pandemic period.

Based on Pearson Chi-square and likelihood ratio test results, there was a statistically significant difference between the yearly distribution and the number of fatal RTAs within each month of accidents (Pearson Chi-square = 68.687, *p* = 0.010 and Likelihood Ratio = 70.200, *p* = 0.007).

### 3.1.3. Age and Gender Variations of Fatal RTA Victims

Comparison of RTAs among gender groups showed that in the 5-year period, the male-to-female ratio was 2.91:1. In the pre-pandemic period, the male-to-female ratio was 2.46:1 in comparison to the pandemic period (4.94:1). We can observe a statistically relevant difference between the number of males involved in fatal RTAs in the pre-pandemic period (*n* = 204, 71.1%) in comparison to the pandemic period (*n* = 90, 83.3%) (Pearson Chi-square = 6.191, *p* = 0.013; likelihood ratio = 6.577, *p* = 0.010). Regarding the female victims, in the pre-pandemic period, 28.9% (*n* = 83) of victims were female, while in the pandemic period, we observed 16.7% (*n* = 18) females.

The age group most prone to fatal RTAs in the 5-year period was 18–50 years (*n* = 202, 51.1%). The same age group is involved in the most fatal RTAs in the pre-pandemic period (*n* = 145, 50.5%) as well as in the pandemic period (*n* = 57, 52.8%). The lowest number of RTA deaths was noted in the underage group (in total *n* = 23, 5.8%; pre-pandemic group *n* = 16, 5.6%; pandemic group *n* = 7, 6.5%). However, regarding the age of the victims, no statistically significant variation was found between the two time periods (Pearson Chi-square value = 2.668, *p* = 0.615).

We further stratified the victims according to age intervals, and we observed that the most affected by RTAs during the pandemic were people in the age group of 31–40 (18.5%), followed closely by age intervals of 41–50 (17.6%) and of 18–30 (16.7%). In comparison, in the pre-pandemic period, the most prone age group was 61–70 (20.5%), followed by the age interval of 18–30 (19.2%). Victims in the age group of 31–40 accounted for 16.1%. The detailed analysis of victims by age intervals regarding the comparison between the two periods is illustrated in Figure 2.

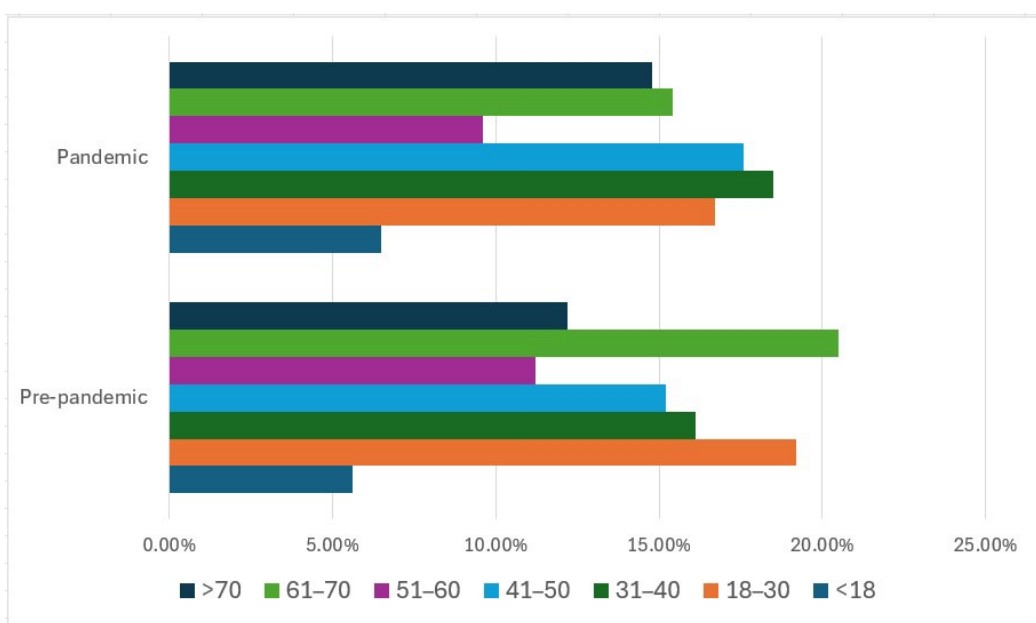

**Figure 2.** Histogram of victims of fatal RTAs according to age groups; comparison of the pre-pandemic period with the pandemic period; a 5-year study (2017–2021) based on medicolegal autopsy records from TILM, Romania.

Further analysis of variations in gender and age of the victims in the two time periods is represented in Figure 3, which shows sex ratios by age for each period.

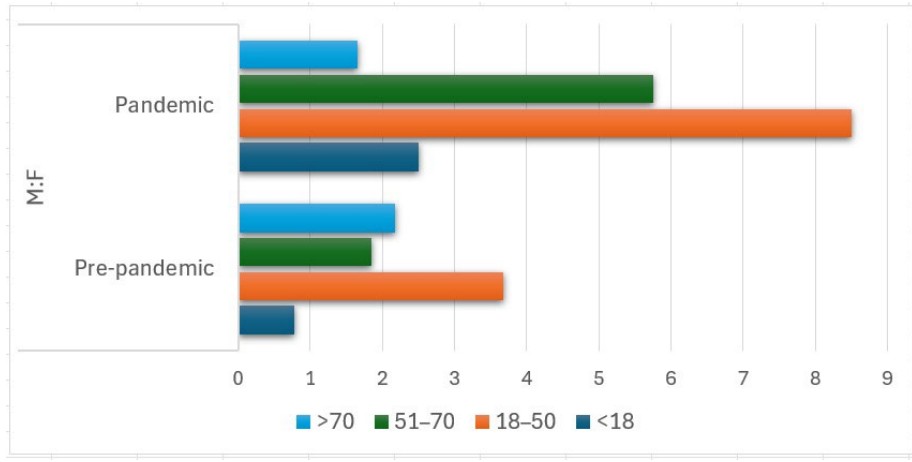

**Figure 3.** Sex ratios by age of fatal RTAs in a 5-year study (2017–2021) based on medicolegal autopsy records from TILM, Romania, pre-pandemic period versus pandemic period.

We observed that males in the age group of 18–50 are the most affected by fatal RTAs in both periods, with no statistically significant difference between their numbers (pre-pandemic $n = 114$, 59.9% versus pandemic $n = 51$, 56.7%; two-proportion Z-test: difference of $-0.0078431$, confidence interval of $(-0.130821, 0.115135)$, Z-value $= -0.13$, $p = 0.901$).

### 3.1.4. Type of Road User

In relation to the kind of road user, in the 5-year period, drivers were the most affected group in fatal RTAs with 121 cases (30.6%). Drivers were most affected by fatal RTAs in both time periods: 26.8% in the pre-pandemic period ($n = 77$) and 40.7% in the pandemic period ($n = 44$). Figure 4 shows the comparison between the frequencies of victims in relation to the kind of road user, matching the two time periods.

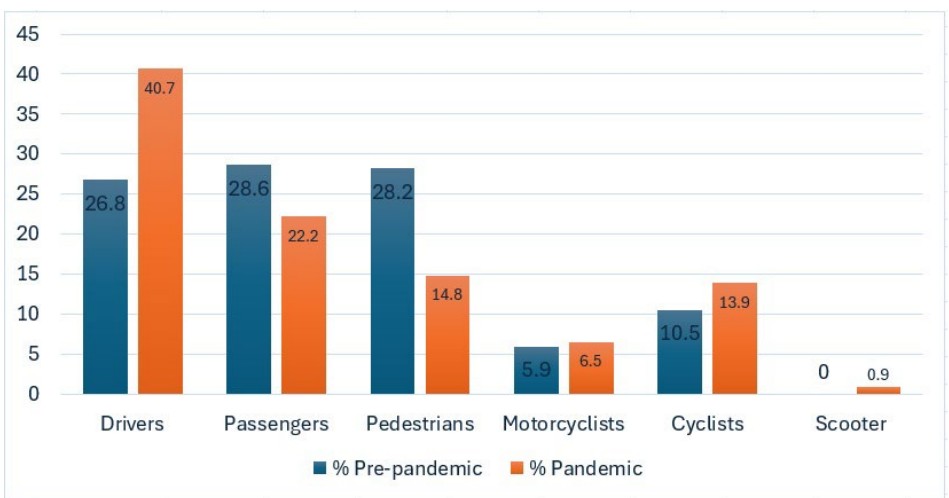

**Figure 4.** Histogram of frequencies of victims in relation to the kind of road user in fatal RTAs by comparing the pre-pandemic period with the pandemic period in a 5-year study (2017–2021) based on medicolegal autopsy records from TILM, Romania.

As illustrated in Figure 4, in both periods, the second most affected group of road users were front- and back-seat passengers, followed closely by pedestrians. However, in the pandemic period, we noted an increase in proportions in comparison to the pre-pandemic period for cyclists (13.9% from 10.5%) and motorcyclists (6.5% from 5.9%). When comparing the two time periods, there is a statistically significant variation in the distribution of victims per kind of road user (Pearson Chi-square = 18.049, $p$ = 0.006 and likelihood ratio = 18.278, $p$ = 0.006). A single case of an electric scooter was discovered in the pandemic period in August 2021.

We observed a notable decrease in the proportion of passengers (from 28.6% to 22.2%) and pedestrians (from 28.2% to 14.8%), and an increase for drivers (from 26.8% to 40.7%), cyclists (from 10.5% to 13.9%), and motorcyclists (from 5.9% to 6.5%) who died in fatal RTAs in the pre-pandemic period in comparison to the pandemic period (Figure 4).

*3.2. Second-Stage Study: Pre-Lockdown Group Versus Lockdown Group Versus Post-Lockdown Group*

The government of Romania enforced restraints and lockdowns during 16 March–14 May 2020 to limit the COVID-19 pandemic. While looking at the lockdown period, it was clear that mobility and RTAs were also affected by the strict regulations that were enforced by the authorities.

In this part of the study, data was further stratified into three groups: the first group was termed the lockdown period (16 March–14 May 2020), the second group was represented by the pre-lockdown period (same period during 2019), and the third group was the post-lockdown period (same period during 2021). We studied the same parameters as in the first-stage study.

Table 2 presents the findings. The sample is small so any statistical analysis was not possible. In the lockdown period, there were 103 medicolegal autopsies performed at TILM, with 8 cases of fatal RTAs (7.76%), while in the pre-lockdown period, the number of total medicolegal autopsies was 122, with 7.38% of them being represented by fatal RTAs. However, in the post-lockdown period, the proportion of fatal RTAs decreased to 6.48% (*n* = 7).

**Table 2.** Comparison among the lockdown period, the pre-lockdown period, and the post-lockdown period, regarding fatal RTAs in a 5-year study (2017–2021) based on medicolegal autopsy records from TILM, Romania.

| | | Pre-Lockdown | | Period Lockdown | | Post-Lockdown | |
|---|---|---|---|---|---|---|---|
| | | Number of Autopsies | Fatal RTAs | Number of Autopsies | Fatal RTAs | Number of Autopsies | Fatal RTAs |
| | | 122 | 9 (7.38%) | 103 | 8 (7.76%) | 103 | 7 (6.48%) |
| Month of the accident | March | | 3 | | 2 | | 2 |
| | April | | 6 | | 5 | | 3 |
| | May | | 0 | | 1 | | 2 |
| Gender of the victim | M | | 8 | | 7 | | 6 |
| | F | | 1 | | 1 | | 1 |
| Type of road user | Driver | | 3 | | 3 | | 3 |
| | Passenger | | 2 | | 1 | | 2 |
| | Pedestrian | | 3 | | 3 | | 0 |
| | Cyclist | | 1 | | 1 | | 1 |
| | Motorcyclist | | 0 | | 0 | | 1 |
| Age of the victim (years) | <18 | | 0 | | 0 | | 0 |
| | 18–50 | | 3 (33.33%) | | 4 (50%) | | 4 (57.14%) |
| | 51–70 | | 4 (44.44%) | | 3 (37.5%) | | 2 (28.57%) |
| | >70 | | 2 (22.22%) | | 1 (12.5%) | | 1 (14.28%) |

## 4. Discussion

RTAs are considered a substantial public health threat and a well-known global problem because of the lives that are lost in such accidents and the impact they have on the population [1]. Seeing that in Romania all victims of fatal RTAs represent medicolegal cases and a medicolegal autopsy is performed, grounded on autopsy results, it is possible to create a comprehensive epidemiological profile that will help in the future in determining the appropriate preventative measures [8,36].

There is a need to draw attention to such catastrophic events and to try to understand the factors that lead to the occurrence of RTAs. With COVID-19 being declared a pandemic on 11 March 2020, national restrictions were regulated, and the mobility and flow of RTAs were affected [13]. Moreover, seeing that Romania held the first place for road deaths among countries in the EU and currently (2023) holds the second place, a need for analyzing road fatalities in this country surfaced. The investigation of risk factors for deadly RTAs is of utmost importance for understanding the dynamics of road accidents.

In the present study, by analyzing victims of fatal RTAs in a 5-year period (2017–2021), based on medicolegal autopsy records from TILM, Romania, we observed a decrease of 44.3% in 2020 in comparison to 2018. The decreasing trend was observed even in the following year (2021), which implies that, due to the COVID-19 pandemic, the number of fatal RTAs was reduced in 2020 and it also carried influence over the next year. These aspects are worth mentioning since they correlate well with national statistics and show that a decreasing trend for RTAs is emerging in Romania.

In Romania, in 2019, 8642 severe RTAs occurred resulting in the death of 1846 people. The number of severe RTAs in 2019 increased by 0.85% from 2018, while the number of fatally injured victims was reduced by 0.34% [38]. Comparing these data with the year of the COVID-19 pandemic, 2020, we see that there is a national decrease in severe RTAs with 27.4% from the preceding year (*n* = 6273) with 1646 of fatally injured victims. Nevertheless, the mortality index (the proportion of deceased victims from all severe RTAs) suffered an increase from 21.6% in 2019 to 26.2 in 2020 [39]. However, in 2021, 4915 severe RTAs happened, associated with 1779 casualties [40]. In 2021, each day of the year led to 13 severe RTAs in which five people died and 10 were badly injured [40]; in comparison, in 2019, 24 severe RTAs happened each day with five lost lives due to injuries resulting from car crashes [38]. In 2020, 17 severe RTAs took place each day, with five daily deaths [39].

A similar study done by Mahendra et al. on road accident tolls throughout the COVID-19 pandemic in Gorontalo City, Indonesia, was consistent with our findings: in 2018, the number of road accidents touched 108 cases (27%); it increased in 2019 with 158 cases (39%); it decreased in 2020 with the number of victims being 87 (21%); that was after Indonesia

began enacting restrictions in 2020 in an effort to slow the coronavirus's rate of transmission and reduce traffic accidents. Then, in 2021, there were fewer traffic accidents and 55 (13%) fewer victims [7]. This was in accordance with a study that analyzed the influence of the lockdown on road deaths in Turkey. In April 2020, road fatalities declined almost by 60% compared to April 2019 [28].

The COVID-19 pandemic restrictions, such as nationwide lockdown and the shutting down of all activities excluding necessary services (all vehicular mobility, construction, and industry work), were the probable cause of the reduction of RTA fatalities [41]. In addition to these restrictions, many adopted personal safety measures out of concern for contracting the virus, like staying indoors and avoiding crowded areas. This population's perspective may also be the cause of the decline in traffic volume and subsequent reduction in the number of RTAs [8].

The lockdown in Spain changed the predicted pattern, with more road fatalities and injuries in 2020 than in the previous two years. Thus, there were fewer accidents overall, particularly the more serious ones, because of the mobility restrictions put in place to stop the COVID-19 virus from spreading [12]. The reduction of fatal RTAs in the lockdown period was explained by Mahendra et al., maintaining that during the pandemic, traffic jams were diminished because people nevertheless spent time at home due to the implemented restrictions, including students, teachers, and company employees [7]. Traffic congestion in main cities was lowered by nearly 15% globally [14]. According to the study by Kamine et al. on the cut in trauma admissions with the COVID-19 pandemic in New Hampshire, America, given the impacts of social distancing, it seems to make sense that the percentage decline in motor-vehicle collision trauma admissions (80.5%) was greater than the percentage decrease in non-motor-vehicle collision trauma admissions (45.1%) [42].

A study by Jefferies et al. that analyzes the effect of the pandemic on major trauma centers in Northern Ireland illustrates a reduction of 26% in admissions in the lockdown period when compared to the same time in 2019 and reduced road deaths in the three months of the lockdown. This was explained by the reduction in traffic volume [27].

However, the one-third decrease in traffic volume did not reduce the number of severe RTAs in Finland, and victims with serious injuries did not fade in lockdown. This indicates that although restrictions were implemented, reliable healthcare services were required for these types of patients, and the pandemic did not modify the incidence of severe injuries [15]. Nonetheless, the same observation was noted by Shimada et al., illustrating that the decreases in traffic on the roads led to an increase in road deaths because of an increase in speeding in some areas [29]. This may explain our findings, the non-relevant difference between the number of victims in the pre-pandemic period and the pandemic period, highlighting the importance of other risk factors that lead to traffic accidents, factors that do not relate to high-volume traffic.

Regarding the association between the yearly distribution and the month of the accident, we noticed a statistically significant difference (Pearson Chi-square = 68.687, $p = 0.010$ and likelihood ratio = 70.200, $p = 0.007$). This is in accordance with conclusions stated by Jiménez-Yanza et al., who analyzed the association of lethality and mortality rates due to COVID-19 and road accidents in Ecuador 2020–2021: from April to June 2020, the number of people who died as a result of road accidents decreased [43].

In our study, when comparing the effects of lockdown on fatal RTAs with the same period during 2019 and 2021, we can see a reduction in the number of cases, but their proportion among all medicolegal cases increased in comparison to the previous year. However, the descending trend is still maintained in 2021. This may be because other situations that require a medicolegal autopsy were reduced, but the number of fatal RTAs in comparison to other circumstances did not reduce enough. Because of the reduced sample of fatal RTAs in the three-time period, a statistical analysis was not possible. An autopsy-based study done by Khurshid et al. on the analysis of RTA fatalities in Karachi, Pakistan in 2019 and 2020 showed that, when the annual spreading of the cases was checked, the overall fatalities described in 2019 (50.4%) were discovered to be like those in 2020 (49.6%).

However, the number of deaths reported in March–June 2020 was 35.6% lower than in March–June 2019, possibly as a result of significant restrictions, including lockdowns, being in place to combat the COVID-19 pandemic in that period [8]. Likewise, a drop in the total count of RTA deaths was seen in Peru during the lockdown period; the research by Calderon-Anyosa et al. showed that, when investigating the initial impact of COVID-19 lockdown on traumatic and accidental fatalities, the major drop was represented by road fatalities, with a decrease of 12.22 deaths per million men monthly (95% CI: 14.45, 9.98) and 3.55 deaths per million women per month (95% CI: −4.81, 2.30) [44].

Our study also showed similar results to research in Greece based on autopsies, which also discovered a noteworthy cut in RTA deaths during the lockdown, which was justified by the decreased overall count of cars in traffic throughout the lockdown [45]. This was also reliably correlative with the findings registered by Saladié et al., who analyzed the impact of the COVID-19 lockdown on the decline in road accidents in Tarragona province, Spain: the number that happened in lockdown dropped by 76% in comparison to 2018–2019 [12].

A study by Siddarth Rao et al., which compared the number of RTAs presented to the Emergency Department before the lockdown and during the lockdown, observed that the number of RTAs reduced by 52.1% compared to a similar period during the previous year [41]. Corresponding to the Head of Traffic at the Gorontalo Police, the overall count of road incidents in Gorontalo City, Indonesia, in January–August 2020 reduced associated with the same period in 2019 (52 cases) [7].

In the present study, more male fatalities occurred in RTAs, with a statistically significant contrast between the pre-pandemic period and the pandemic period ($p = 0.013$), showing an increase in male victims in the latter. This is in accordance with studies by Siddharth Rao et al. (1.3:1 male-to-female ratio in the pre-lockdown period and 3:1 in the lockdown period) and Khurshid et al. (6.03:1 [211 (85.8%) males and 35 (14.2%) females]) [8,41].

The most affected age group was 18–50 years, with similar proportions in both periods (50.5% and 52.8%). This suggests that productive age groups are more vulnerable to fatal RTAs, even more so in the pandemic period. The study conducted by Siddharth et al. showed consistent results with our study; 16–45 years was the most common age group both in the pre-lockdown and lockdown period [41]. Similar results were obtained by Khurshid et al.; the highest number of deaths was noted in victims aged between 18 and 40 years ($n = 134$, 54.5%), whereas the minimal number was in victims aged ≥60 years (8.5%) [8]. However, in our study, the lowest number of victims was noted in the age group of <18 years. Although the number of underage people who die in fatal RTAs has reduced with the COVID-19 pandemic, their proportion remains similar (5.6% in the pre-pandemic period and 6.5% in the pandemic period), with a slight increase during the pandemic. This suggests that there is a need to increase awareness of the dangers of RTAs for this age group.

With further stratification of the victims according to age intervals, we see that people in the age group of 31–40 (18.5%), followed closely by age intervals of 41–50 (17.6%) and 18–30 (16.7%), were the most affected by the pandemic. However, in the pre-pandemic period, the most prone age group was 61–70 (20.5%), followed by the age intervals of 18–30 (19.2%) and 31–40 (16.1%). It is suggested that young people were not protected by the pandemic. Moreover, people over 60, maybe due to reduced mobility and installed curfews or because of voluntarily opting for a stay-at-home strategy out of concern for their own health, were less involved in the pandemic. It appears that the age group of >70 years was more involved in fatal RTAs in the pandemic period (14.8%) than in pre-pandemic (12.2%), however, with no statistical relevance. This aligns with the research conducted by Sakelliadis et al., which indicated a growth in the age group of 70–79 years in 2020 [45]. In Finland, as reported by Riuttanen et al., contrary to expectations of a decrease in the age profile of injury victims, the average age of badly wounded patients rose from 47 years to 53 years, with 30% of patients above 70 years [15].

The most RTA deaths appeared in males aged 18–50 years. A study conducted in Karachi, Pakistan, showed consistent results with our study, with males aged 18–40 years being more prone to fatal RTAs (49.2%) [8].

In relation to the kind of road user, drivers were most prone to fatal RTAs in both time periods. This agrees with the study by Jiménez-Yanza et al. that illustrated that drivers account for the greatest number of fatalities in accidents. Hence, there is a need for thorough vehicle examination to verify the presence of safety devices and technical breakdowns that would assist with the avoidance of fatal consequences [43].

In our research, the second most affected type of road user was represented by passengers, followed by pedestrians, with a decrease in the pandemic period. This phenomenon could be explained by the fact that, in addition to restrictions such as the closing of nonessential services, people took personal protective measures, avoiding gatherings out of fear of developing the disease [8,41]. According to a study by Chiba et al., which looked at how the lockdown affected the epidemiology and outcomes of trauma admissions at the major trauma hospital in Los Angeles, there was a 42.5% drop in the number of auto vs. pedestrian admissions during the lockdown [46]. The original research by Redelmeier D.A. and Zipursky J.S. on pedestrian fatalities during the COVID-19 pandemic found a 50–60% cut in pedestrian activity during the COVID-19 lockdown in New York, with 3.8 pedestrian deaths per month in comparison to 9.7 per month in the three prior years; the inconsistency showed a 60% relative reduction in total pedestrian fatalities (95% confidence interval 34 to 46, $p < 0.001$). In Toronto, 0.5 pedestrian fatalities happened per month during lockdown in comparison to 3.1 in the three prior years (84% relative drop in total pedestrian fatalities, 95% confidence interval 30 to 94, $p < 0.001$) [47].

With reference to other road users, in the two time periods, the proportions were similar, observing a slight increase in the pandemic for both cyclists and motorcyclists. The review by Yasin et al. on the overall effect of the COVID-19 pandemic on road accidents showed that road fatalities increased in the Czech Republic during the lockdown by 86% for cyclists and 50% for motorcyclists [14]. In a study comparing traffic accident victims who came to the Emergency Room of a teaching hospital before and after lockdown, Siddharth Rao et al. found that bike skid was the most common mode both before and during lockdown, with a notable increase of 28.6% during the lockdown period [41]. In Australia, there was a cut in the number of road fatalities among pedestrians (by 20%), motorcyclists (by 12%), passengers (by 11%), and drivers (by 5%), but cyclist fatality increased by 29% during lockdown compared with the prior three years [14]. Bicycle trauma increased during the pandemic, as evidenced by a study that was included in the meta-analysis conducted by Kabiri et al. on the effect of the COVID-19 pandemic on hospital admissions resulting from traffic crashes. People rode bicycles for transportation [37]. Nonetheless, this phenomenon is in contrast with a study by Chiba et al. on the influence of the lockdown on the epidemiology and consequences of trauma admission in Los Angeles, which shows that bicycle-injury admissions were reduced by 28.4% [46].

Regarding our study, a limitation that needs to be noted is the fact that in the second-stage study, the number of victims is very limited, and no statistical tests could be applied, in comparison to other relevant studies with a larger sample which strictly analyzed the effect of lockdown on fatal RTAs. Moreover, the number of subjects included in the study ($n = 395$) is limited, and the results apply to the western part of Romania, but they fit well with the mentioned literature. It is also important to note that while some chose to stay at home out of worry for their own and their loved ones' health, we should not solely blame restrictive measures for any declines in accident rates.

To summarize, although the restrictive measures imposed by the Romanian government led to a reduction in vehicular and human mobility, the number of road fatalities was not significantly reduced in the pandemic period in comparison to the pre-pandemic period. This means that in addition to the high-volume traffic that leads to RTAs in Romania, a cumulation of other risk factors needs to be addressed. According to the literature, factors related to the driver are the most important ones, since careless/reckless driving is the most

frequent. Driving while tired, impaired by alcohol or other drugs, disregarding the road, over-speeding, and disregarding the traffic laws represent recipes for disaster, as illustrated in our previously published article [10].

## 5. Conclusions

Our findings show that measures implemented to control the COVID-19 pandemic and the spread of the virus by limiting traffic of any kind may have had a positive effect on the reduction of traffic accidents, as shown by the information based on medicolegal autopsies in Timis County, Romania, but more attention needs to be paid to other risk factors. The novelty of the study consists in adding Romania to the vast list of countries that were discussed regarding the COVID-19 pandemic's influence on RTAs. Moreover, since Romania was the leading country in the EU in road deaths, this study is of maximum importance to assess whether the restrictive measures imposed by the government helped reduce road fatalities. However, the reduction factor was not statistically significant, hence the importance of other risk factors that lead to road fatalities needing to be addressed. Additional research is required to determine the causes of the slight decrease in fatal injuries that occurred when traffic volume was decreased during a mandatory nationwide lockdown.

In addition, the most important findings of this research were that there was a statistical significance between the monthly distribution of cases regarding the year of the accident and the most affected age group was 18–50 years old, but there was no statistical significance between the number of victims in the pre-pandemic period and the pandemic period. With further analysis of the victims according to age intervals, we observed changes in the age of the victims in the two time periods: in the pandemic period, the most affected age groups were 31–40, 41–50, and 18–30 years old, in this order. In the pre-pandemic period, however, the most affected age group was 60–70 years, followed closely by 18–30 years old, suggesting that the pandemic and the imposed restrictions modified the age of the victims, protecting older people. However, we do acknowledge that COVID-19 restrictions did not protect the active age group, and these victims need more protective measures and information campaigns directed to them. Male victims were the most affected, which is an aspect we saw in both periods. This is in accordance with the literature, since male victims are more affected by fatal RTAs. Regarding the types of road users, drivers were the most affected group, followed by passengers and pedestrians, with no significant difference between the two time periods that we analyzed. We observed a decrease in the number of passengers and pedestrians involved in fatal RTAs when comparing the pandemic period with the pre-pandemic period and a slight increase for cyclists and motorcyclists, with no statistically significant difference. However, there was an increase in the proportion of drivers in the pandemic period, which suggests that other types of road users may have been protected by the restrictions that were implemented but with no protection for drivers.

Future directions and perspectives for us on the subject of fatal RTAs include the analysis of risk factors that lead to road deaths, such as alcohol and drug consumption among Romanian drivers, and the development of pattern injuries derived from medicolegal autopsy records since we did not analyze in this current study the traumatic injuries and the cause of death for the subjects.

Once again, this is evidence of the disruptive effect of the COVID-19 pandemic on road transport and human and vehicular mobility and how such a global spread of an infectious agent can influence the frequency of accidents and the number of victims involved. Lessons learned from the COVID-19 pandemic illustrate that addressing vehicular movement by perfecting road infrastructure, having more police surveys, and more rigorous travel rules, may lead to reduced traffic collisions and road deaths. These can be addressed in future policy developments towards road safety improvements in case of unique events such as an infectious pandemic. Understanding various aspects of road safety and driving behavior in case of road fatalities when less traffic congestion is present represents the goal

for preventative campaigns targeted at reducing road deaths in such unique circumstances. This can help implement dedicated programs during governmentally imposed lockdowns.

**Author Contributions:** Conceptualization, Ş.U. and A.E.; methodology, Ş.U. and V.C.; software, C.P.; validation, V.C., C.-O.M. and A.E.; formal analysis, Ş.U., V.C. and A.E.; investigation, Ş.U., E.S., G.-D.G. and A.S.; resources, Ş.U., E.S., G.-D.G. and A.S.; data curation, C.P.; writing—original draft preparation, Ş.U., E.S., G.-D.G. and C.-O.M.; writing—review and editing, Ş.U., V.C. and A.E.; visualization, Ş.U., C.-O.M. and A.E.; supervision, A.E. All authors have read and agreed to the published version of the manuscript.

**Funding:** This research received no external funding.

**Institutional Review Board Statement:** The study was conducted in accordance with the Declaration of Helsinki and was approved by the Research Ethics Committee of Victor Babes University of Medicine and Pharmacy Timisoara, registered under decision number 88/19.12.2022 rev 2024.

**Informed Consent Statement:** Not applicable.

**Data Availability Statement:** The data presented in this study are available on request from the corresponding author. The data are not publicly available due to confidentiality reasons.

**Acknowledgments:** We would like to acknowledge Victor Babes University of Medicine and Pharmacy Timisoara for their support in covering the costs of publication for this research paper.

**Conflicts of Interest:** The authors declare no conflicts of interest.

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
