# Peer review of "The Impact of the COVID-19 Pandemic on Fatal Road-Traffic Accidents: A Five-Year Study on Medicolegal Autopsies in Timis County, Romania"

_safety_

Round 1
Reviewer 1 Report
Comments and Suggestions for Authors
See attached document.

Reviewer 2 Report
Comments and Suggestions for Authors
This paper does not really show new findings.
The conclusion is quite strange, why would the effect of a pandemic be a first step "in promoting preventive strategies based on restrictive policies"?
Reviewer 3 Report
Comments and Suggestions for Authors
The field of study is an interesting one, but it should be mentioned that from a statistical point of view for someone who is also active in the field of research in the extended area of ​​vehicular road communications, safety systems and the management of ITS systems in view of reducing the degree of death among drivers, pedestrians or passengers, it is normal that the measures imposed during the pandemic reduce the number of deaths.
However, we are talking about over 1.3 million deaths worldwide, over 50 million victims, and the list can go on. It would have been interesting if the following aspect emerged from the study, in which a person who died as a result of an accident and was infected with Sars-Cov2 would have had more chances of survival if they were not infected. Taking into account the mental state, the lucidity, the symptoms that an infected person has. Here it is only a purely personal opinion by which to identify an extrapolation of the results with an extensive impact.
Observations
Chapter 1 is ok, maybe a wider exposition with clearer examples of other research groups would have been indicated so that there is a comparative terminology, but otherwise it seems ok.
Chapter 2 must serve a methodology, a way of presenting what was used, what other examples of articles were the basis of the study. If there were formulas, methods, materials used, this is not clearly visible at the moment. Please improve.
Chapter 3 is a complete and complex one as an effort of some analyses, but what we want is to outline a model based on which we can analyze both in time and in efficiency and to deduce as a result of the model some characteristics by applying which we have a case reduction factor.
The exposed data are interesting and their analysis is as comprehensive as possible, especially since it comes from people with access to medical information.
Can you generate a mathematical model to base the analyzes on?
What should we understand as a result of these results presented in chapter 3?
How does this study improve the specialized literature?
The discussions in chapter 4 would ideally be based on the results obtained and their exposure in a comparative way with other studies carried out and what differentiates what you did and what they did.
What are the elements of novelty and the contribution brought by this article?
How do you consider specialized literature to help your study and how can this direction be continued?
Can you explain to us what are the clear directions you have in this field and how do you improve what was worked on in the current manuscript?
All this information that reflects the novelty, the contribution, the future directions, should be found in the conclusions.
Regarding the references, you can try to adjust their number and find other impact studies to add because many of the added links, especially those police releases do not lead to an address, this is incomplete.
Reviewer 4 Report
Comments and Suggestions for Authors
According to the paper's title, the authors assess the Impact of the C19 Pandemic on the incidence and characteristics of fatal road accidents in a specific region of Romania. Looking at the "Safety" journal profile, I can say that the proposed article fits with that profile (road safety).
The article's structure is clear and does not raise any objections. A review of knowledge is presented, followed by formulation of the research problem. Then, the research method results and their analysis are presented. The article is summarized with conclusions.
I have no major substantive comments. The authors present interesting research results on the impact of the COVID-19 pandemic in a specific area of human activity and safety – road safety. However, it should be remembered that the research sample is limited, which makes it difficult to generalize conclusions. It is essential in this context to refer to other similar analyses.
Other remarks:
Line 43-48: The quoted WHO data refer to 2016. According to the latest data published by this organization, road accident fatalities have already “dropped out” of the Top 10 causes of death overall some time ago. It is worth verifying this data.
The drawings should be made in better resolution (some labels are illegible).
Section Conclusion: It is worth repeating the most essential specific quantitative and qualitative conclusions from the conducted analyses rather than only limiting ourselves to generalities.
Round 2
Reviewer 1 Report
Comments and Suggestions for Authors
No comment.
Author Response
Dear Reviewer 1,
Thank you very much for your approval of the manuscript. We very much appreciate and value your expertise.
King regards,
The Authors
Reviewer 2 Report
Comments and Suggestions for Authors
Thank you for the vast number of extensions of the text. The paper improved a lot.
Author Response
Dear Reviewer 2,
Thank you very much for your feedback. We are pleased we managed to make changes that improved the manuscript. We appreciate and value your opinion and expertise and we are glad for your favorable assessment.
Kind regards,
The authors
Reviewer 3 Report
Comments and Suggestions for Authors
There are no other aspects to correct. What was initially transmitted was respected by the authors.
There is a mention regarding the percentage of similarity that I observe in the report, and it exceeds 30%. Here we ask you to adjust those paragraphs or chapters so that the degree of similarity is one agreed by the journal.
Author Response
Dear Reviewer 3,
Thank you very much for your comments. We highly appreciate your opinion and expertise.
Comment 1: There are no other aspects to correct. What was initially transmitted was respected by the authors.
Response 1: Thank you for your opinion. We are glad you found the revised version of the manuscript with the right corrections. We value your suggestions and we thank you for your observations and for the changes we made according to them.
Comment 2: There is a mention regarding the percentage of similarity that I observe in the report, and it exceeds 30%. Here we ask you to adjust those paragraphs or chapters so that the degree of similarity is one agreed by the journal.
Response 2: Thank you for your observation. We have made changes in the manuscript addressing the similarity percentage. We hope the degree of similarity has reduced to be favorable to the journal.
Kind regards,
The Authors
Reviewer 4 Report
Comments and Suggestions for Authors
The authors responded to the submitted comments substantively and introduced appropriate corrections.
Author Response
Dear Reviewer 4,
Thank you for your feeding. We are glad you found the changes we made to be appropriate. We value your opinion and we thank you for your assessment.
Kind regards,
The authors
Round 3
Reviewer 3 Report
Comments and Suggestions for Authors
The article has improved substantially and there are no other comments. The authors have respected every aspect submitted.